# The Influence of Warm-Up on Body Temperature and Strength Performance in Brazilian National-Level Paralympic Powerlifting Athletes

**DOI:** 10.3390/medicina56100538

**Published:** 2020-10-14

**Authors:** Marcelo de Aquino Resende, Roberta Barreto Vasconcelos Resende, Gracielle Costa Reis, Layanne de Oliveira Barros, Madson Rodrigo Silva Bezerra, Dihogo Gama de Matos, Anderson Carlos Marçal, Paulo Francisco de Almeida-Neto, Breno Guilherme de Araújo Tinoco Cabral, Henrique P. Neiva, Daniel A. Marinho, Mário C. Marques, Victor Machado Reis, Nuno Domingos Garrido, Felipe J. Aidar

**Affiliations:** 1Department of Physical Education, Tiradentes University (UNIT), Aracaju, Sergipe 49010-390, Brazil; roberta.resende@souunit.com.br (R.B.V.R.); gracielle.reis@souunit.com.br (G.C.R.); madson.rodrigo@souunit.com.br (M.R.S.B.); 2Group of Studies and Research of Performance, Sport, Health and Paralympic Sports (GEPEPS), Federal University of Sergipe (UFS), São Cristovão, Sergipe 49100-000, Brazil; layanneoliveira@cepiexpansao.com.br (L.d.O.B.); dihogogmc@hotmail.com (D.G.d.M.); acmarcal@academico.ufs.br (A.C.M.); fjaidar@academico.ufs.br (F.J.A.); 3Program of Physical Education, Federal University of Sergipe (UFS), São Cristovão, Sergipe 49100-000, Brazil; 4Department of Physical Education, Federal University of Rio Grande do Norte (UFRN), Natal, Rio Grande do Norte 59078-970, Brazil; paulo220911@hotmail.com (P.F.d.A.-N.); brenotcabral@gmail.com (B.G.d.A.T.C.); 5Department of Sport Sciences, University of Beira Interior, 6201-001 Covilhã, Portugal; henriquepn@gmail.com (H.P.N.); dmarinho@ubi.pt (D.A.M.); mmarques@ubi.pt (M.C.M.); 6Research Center in Sports Sciences, Health Sciences and Human Development (CIDESD), Trás os Montes and Alto Douro University, 5001-801 Vila Real, Portugal; victormachadoreis@gmail.com; 7Department of Physical Education, Federal University of Sergipe (UFS), São Cristovão, Sergipe 49100-000, Brazil; 8Program of Physiological Science, Federal University of Sergipe (UFS), São Cristovão, Sergipe 49100-000, Brazil

**Keywords:** warm-up, powerlifting Paralympic, athletes, strength, temperature

## Abstract

*Background and Objectives:* The effects of warm-up in athletic success have gained strong attention in recent studies. There is, however, a wide gap in awareness of the warm-up process to be followed, especially in Paralympic powerlifting (PP) athletes. This study aimed to analyze different types of warm-up on the physical performance of PP athletes. *Materials and Methods:* The sample consisted of 12 elite Brazilian PP male athletes (age, 24.14 ± 6.21 years; bodyweight, 81.67 ± 17.36 kg). The athletes performed maximum isometric force (MIF), rate of force development (RFD), and speed test (Vmax) in three different methods of warm-up. Tympanic temperature was used to estimate the central body temperature. *Results:* A significant difference was observed for MIF in the without warm-up (WW) condition in relation to the traditional warm-up (TW) and stretching warm-up (SW) (*p* = 0.005, η^2^_p_ = 0.454, high effect). On the contrary, no significant differences were observed in RFD, fatigue index (FI) and time in the different types of warm up (*p* > 0.05). Furthermore, no significant differences were observed in relation to the maximum repetition (*p* = 0.121, η^2^_p_ = 0.275, medium effect) or the maximum speed (*p* = 0.712, η^2^_p_ = 0.033, low effect) between the different types of warm up. In relation to temperature, significant differences were found for the TW in relation to the “before” and “after” conditions. In addition, differences were found between WW in the “after” condition and SW. In addition, WW demonstrated a significant difference in relation to TW in the “10 min later” condition (F = 26.87, *p* = 0.05, η^2^_p_ = 0.710, high effect). *Conclusions:* The different types of warm-up methods did not seem to provide significant differences in the force indicators in elite PP athletes.

## 1. Introduction

Warm-up routines are a standard practice before training and competition in most sports. Coaches recommend warm-ups to avoid injuries and boost their athletes’ performance [1,2]. Furthermore, the warm-up has been an important topic of research and previous studies have provided strong evidence of its effectiveness [3,4]. A well-designed warm-up appears to induce physiological changes and enable the athlete to improve mental concentration for the next challenge [5].

Studies have described physiological adaptations to warm-up that theoretically support a positive effect of warm-up on subsequent performance [6,7]. These effects are mostly associated with an increase in body temperature [6,7]. It has also been proposed that the rise in muscle temperature caused by priming exercises contributes to various physiological and metabolic changes, influencing efficiency [4]. Nevertheless, questions have been raised about the efficacy of the warm-up in increasing athletic performance and avoiding injury [8]. Warming up can also help athletes in team sports by preparing them for the skills they need in basketball, handball, and baseball [8,9].

The numerous parameters and the nature of their partnership make it difficult to characterize the key features of the correct warm-up technique [3]. Therefore, more evidence is required to test the efficacy of the warm-up to minimize muscle injury, and improve athletic performance, which should be performed considering the need for new approaches that contribute to better performance.

Among the several warm-up procedures in the past few years, researchers have been focusing on the effects of post-activation performance enhancement, suggesting that it may improve aspects of muscle power [10,11]. This type of warm-up–induced increase in the production of force typically occurs after a maximum or close maximum stimulation of the muscle [12]. In addition, recent investigations have been indicating that stretching performed before exercise, particularly if it lasts longer than 60 s, can reduce muscle strength [13,14]. Stretch-induced output declines are especially apparent in the maximal and explosive physical activity that plays a central role in a number of sports. Nevertheless, stretching during a sporting event is often meant to minimize muscle fatigue or improve muscle compliance, thus potentially reducing the possibility of an injury [15]. Thus, it is important to study the techniques that optimize the positive benefits of stretching and reduce the negative effects (reduction in muscle performance).

On the other hand, some studies involving Paralympic athletes have focused on the etiology of preventing possible injuries and classification criteria in competition [16]. In addition, our group recently analyzed the effect of different forms of post-exercise recovery involving non-steroidal anti-inflammatory drugs [17] and creatine supplementation [18]. Therefore, due to the lack and growing interest in studies in this population, it is necessary to understand the influence of warm-up in this segment.

In this study, we analyzed the effect of different types of warm-up on the physical performance of elite Paralympic powerlifting (PP) athletes. We hypothesized that warming up before training may boost the performance of athletes.

## 2. Materials and Methods

### 2.1. Sample

The study sample comprised 12 male PP athletes participating in the extension project of the Federal University of Sergipe, Sergipe, Brazil. All the athletes were nationally classified competitors, eligible to compete in the sport [19]. As an inclusion criterion, the athlete had to have participated in at least one competition at the national level in the period of 1 year and have a minimum experience of 1 year in the sport. Four athletes had spinal cord injuries due to accidents with injuries below the eighth thoracic vertebra, four had amputations, two had polio, one had cerebral palsy, and one had artogryposis. The participant characterization as well as the data collected in Session 1 of the first week is presented in Table 1.

The athletes participated voluntarily and signed an informed consent form in accordance with the Resolution 466/2012 of the National Commission for Research Ethics (CONEP) of the National Health Council and the ethical principles of the latest version of the Declaration of Helsinki 2013 (and the World Medical Association). This study was approved by the Research Ethics Committee of the Federal University of Sergipe, CAAE: 2.637.882 (date of approval: 7 May 2018).

### 2.2. Experimental Design

In this study, we analyzed, through static and dynamic strength tests the effects of three different types of warm-up and the effect of each method on the strength of national-level PP athletes. All procedures were performed at the same time of day for and under the same environmental conditions (between 23 °C and 25 °C of temperature and relative humidity of ~60%). Following familiarization, each participant completed the strength tests on the adapted bench press [19]. The study was conducted over 3 weeks.

In week 1 the athletes all performed 3 familiarization sessions of the following tests: 1 maximum repetition (1RM), maximum isometric force (MIF) test with the bar at 15 cm from the chest, rate of force development (RFD), fatigue index (FI), time to maximum isometric force (time), and maximum speed (Vmax). In Weeks 2 and 3, the athletes were randomly assigned, by drawing lots, to one of three warm-up forms; without warm-up [WW], traditional warm-up [TW], or stretching warm-up [SW]), with one third of the participants performing each type of warm-up in one of the evaluation sessions. At the beginning of testing in Weeks 2 and 3, each participant remained seated for 5 min for the heart rate (Vantage NV; Polar, Kempele, Finland) and temperature measurements. After performing one of the three types of warm-up, the participants remained at rest for 10 min [21], and after that, they performed the force tests. The athletes were asked to maintain the same routine during the evaluation days, avoiding strenuous exercise and refraining from consuming caffeine 48 h before the test. The second week was dedicated to the 1RM and Vmax tests. The third week was dedicated to measuring static force variables. A summary of the design is presented in Figure 1.

### 2.3. Methods

The bodyweight of the athletes was measured using a Michetti digital electronic scale, Model Mic Welchair (Michetti, Brazil), to facilitate weighing that is usually done in a sitting position, with a maximum supported weight capacity of 500 kg and a dimension of 1.02 × 1.20 m^2^. To perform the bench press, an official straight bench (Eleiko, Sweden), approved by the International Paralympic Committee [19], measuring 210 cm in total length, was used. The bar used was the 220 cm Eleiko brand (Eleiko, Sweden) weighing 20 kg [19].

### 2.4. Temperature

Tympanic temperature was used to estimate the central body temperature (Braun Thermoscan IRT 4520, Kronberg, Germany) [22,23]. After the athletes remained seated for 5 min before starting the tests, the temperature was checked 3 times and the average value was used. The same procedure was performed before warm-up and 10 min after warm up, in the three methods of warm up [24,25].

### 2.5. Determination of Load (1RM) and Maximum Speed (Vmax)

The athletes performed the trials with a weight that they estimated could be lifted once using the maximum effort. Weight was then added until the maximum load that could be lifted with one repetition was reached. If the participant was unable to perform a single repetition, 2.4–2.5% of the load used in the test was subtracted [26]. A 3- to 5-min rest was provided between attempts. All participants performed the test before and after training, with a minimum interval of 10 min between the tests and the training session.

To measure the speed of movement, a valid and reliable linear position transducer [27], Model Speed 4Lift (Speed4Lift SL Force Measurement System; Mostoles, Madrid, Spain) was attached to the bar. The maximum speed averages were collected for analysis purposes with loads of 100% 1RM [28].

### 2.6. Isometric Force Measurements

The measures of muscle strength, RFD (N·s^−1^), MIF (N), FI (%), and time to MIF (s), were determined by a Chronojump force sensor (Chronojump, BoscoSystem, Madrid, Spain) with capacity of 500 kg, output impedance of 350 ± 3 ohm, insulation resistance >2000 cc, input impedance 365 ± 5 ohm, and digital analog converter 24-bit and 80 Hz. The equipment was attached to the adapted bench, using Spider HMS Simond carabiners (Chamonix, France), with a breaking load of 21 KN, approved for scale by the Union Internationale des Associations d’Alpinisme (UIAA). A steel chain with a breaking load of 2300 kg was used to secure the force sensor to the seat. The perpendicular distance between the force sensor and the center of the joint was determined and used to calculate joint torques and FI [29]. MIF was measured by the maximum isometric force generated by the muscles of the upper limbs. The MIF was determined in Newton (N) conceived by the formula N = (M) × (C), where M = mass in kilogram and C = 9.80665, measured between the force sensor cable fixation point and the seat of the bench press, which was adjusted such that there was an angle of the elbow close to 90°, and at a distance of 15 cm of the chest. For the MIF assessment, participants were instructed to perform a single maximum movement looking for elbow extension (as quickly as possible) and to relax. For the fatigue index (FI) assessment, the same exercise was performed and determined that the participants maintain the maximum contraction for 5.0 s, where the index was measured by dividing the initial MIF in relation to the final MIF, subtracted from one. FI = (maximum force − minimum force/maximum force) × 100. The RFD was determined using the force-to-time ratio until reaching the maximum force (RFD = Δforce/Δtime) in 300 ms. The time to maximum force was determined in time to reach the MIF [30].

### 2.7. Warm-Up Protocol

#### 2.7.1. Without Warm-Up

In the WW procedure, the participants did not perform any type of specific warm-up and remained at rest until performing the tests.

#### 2.7.2. Traditional Warm-Up

The participants performed previous warm-up for the upper limbs, using three exercises (abduction of the shoulders with dumbbells, military press with dumbbells, and medial and lateral rotation of the arm to warm up the rotator cuff with dumbbells) with a set of 20 repetitions in approximately 10 min. Subsequently, specific warm-up was performed on the bench press using only the barbell (20 kg) without extra weight, with 10 slow repetitions (3.0 × 1.0 s, eccentric × concentric) and 10 fast repetitions (1.0 × 1.0 s, eccentric × concentric). Next, the participants performed five repetitions with 30% of 1RM, followed by three repetitions with 50% of 1RM, a repetition with 70% of 1RM, a repetition with 80% of 1RM, and a repetition with 90% of 1RM. Between the series, the participants rested for 5 min [31].

#### 2.7.3. Stretching Warm-up

The participants performed only three static stretching exercises for deltoids, chest, and triceps, as is shown in Figure 2 [32]. The exercises were repeated 3 times, with an interval of 10 s [32].

### 2.8. Statistical Analysis

Descriptive statistics was performed using the measures of central tendency, mean (X) ± standard deviation (SD), and 95% confidence interval (95% CI). To verify the normality of the variables, the Shapiro–Wilk test was used in view of the sample size. The data for all variables analyzed were homogeneous and normally distributed. To assess the temperature of the conditions and moments, the two-way ANOVA test, with Bonferroni’s post hoc correction, was performed. The one-way ANOVA test, with Bonferroni’s post hoc correction, was performed to evaluate the differences between the types of warm up. All statistical analyses were performed using the computerized package Statistical Package for the Social Science (SPSS), version 22.0. The level of significance was set at *p* < 0.05. To check the effect size partial Eta squared (η^2^_p_) was used, adopting values of low effect (≤0.05), medium effect (0.05 to 0.25), high effect (0.25 to 0.50), and very high effect (>0.50) for ANOVA [33].

## 3. Results

The results for the studied variables (RFD, MIF, FI, USA, and time to maximum isometric force) are presented in Table 2. A significant difference was observed only for the MIF in the WW condition in relation to the TW and stretching conditions. For the RFD, FI, and time, no significant differences were observed in relation to the warm up conditions.

The results for the 1RM and maximum speed at different methods of warm-up are presented in Figure 3. Figure 3 shows the results found for maximum repetition (kg) and maximum speed (m·s^−1^).

No significant differences were found in relation to the maximum repetition (*p* = 0.121, η^2^_p_ = 0.275, medium effect) or the maximum speed (*p* = 0.712, η^2^_p_ = 0.033, low effect) between the different types of warm up.

The results for the tympanic temperature at different times of measurement are presented in Figure 4.

Significant differences were found between the moments in the traditional warm-up groups at the moments (before the tests) (36.52 ± 0.21 °C, 95% CI 36.39–36.66) and after the test (36.58 ± 0.19 °C, 95% CI 36.46–36.70) “a” *p* = 0.035. It was observed that, WW between the moments after the test (36.47 ± 0.12 °C, 95% CI 36.39–36.54) and the moment 10 min after the test (36.40 ± 0.10 °C, 95% CI 36.33–36.47), with “b” *p* = 0.002. A difference was found at the moment after the test, between WW (36.47 ± 0.12 °C, 95% CI 36.39–36.54), and TW (36.58 ± 0.19, 95% CI 36.46–36.70), “c” *p* = 0.007, and between WW and SW (36.55 ± 0.19 °C, 95% CI 36.43–36.67), “d” *p* = 0.031. At the moment 10 min after, differences were found between WW (36.40 ± 0.10 °C, 95% CI 36.33–36.47) and TW (36.62 ± 0.11 °C, 36.55–36.69), “e” *p* < 0.001, F = 26.87, η^2^_p_ = 0.710 (high effect).

## 4. Discussion

The present study aimed to verify the effect of performing different warm-up protocols on the possibility of improving the performance of PP athletes. The main findings indicated the following: (1) There was a difference only for MIF in the WW condition in relation to the other warm-up conditions. (2) There were no significant differences in the RFD, FI, and time in the different types of warm-up. (3) No differences were found in relation to the maximum repetition or in relation to the maximum speed between the different types of warm-up. (4) With regard to the temperature, differences were observed between the different moments evaluated.

With regard to the isometric mechanical indicators, it was not possible to observe significant differences after performing the three types of warm-ups in the FI. Tsoukos et al. [34] observed that, after the previous warm-up on a cycle ergometer and dynamic stretching, for performing bench press exercise, it can cause excessive fatigue and result in a decrease in performance [34]. In the same direction, the capacity for muscle contraction is directly affected by the muscle work performed previously, with residual effects from previous activities [35]. In addition, Raddi et al. [36] demonstrated that previous warming was not able to improve the production of upper limb strength. Our findings corroborates with those results. In the present study, there were also no significant differences in the RFD in relation to the different warm-up methods. Young and Behm [37] observed that different warm-up methods tend to reduce the performance of jumps.

With regard to the MIF, the results of the present study demonstrated that there was a greater generation of force in the WW when compared to the SW. In addition, in another research performed by our group [32], stretching was found to be a factor that interfered with peak torque. That is, the longer the time after stretching, the less effect will be observed on the isometric force.

With regard to time to maximum force, no differences were found between the warm-up methods. Thus, it seems that the kinetic variables were not affected by the type of warm-up. Trajano et al. [38] found a drop in isometric force and contractile capacity after a 5-min stretching intervention, with recovery only after 15 min. In our study, we adopted a 10-min recovery, which could explain why there were no differences in the isometric force indicators between warm-ups. It is worth mentioning that the present study analyzed Paralympic powerlifting athletes while the study by Trajano et al. [38], analyzed healthy non-athlete men. This may suggest that the trainability of the samples from both studies may have influenced the results.

Regarding the dynamic force variables, no differences were observed between the different types of warm-up. In line with our results, Molacek et al. [39] reported that, regardless of the type of warm-up or stretching performed on the bench press, there are no significant differences in muscle strength variables.

We noticed that the highest temperatures occurred mainly in the TW and SW methods (after the test and 10 min later). When assessing the warm-up and its relationship to body temperature, the main objective of warm-up exercises is to increase the body temperature. The increase in central and muscular temperature alters several physiological functions that would improve physical performance [6], which supports our findings of an even, if not significant, increase in tympanic temperature.

In this context, some physiological changes caused by warm up are expected, such as increased blood perfusion and oxygen consumption, which would lead the athlete to anticipate the beginning of the steady state, allowing for the beginning of more intense activities with greater safety [40]. In addition, the increase in muscle temperature tends to decrease joint stiffness and improve the action of muscle fibers during exercise, providing more coordinated and faster activities [6]. Thus, the increase in body temperature may be the only mechanisms that could help improve physical performance, and that is usually achieved through warming up [6,40].

However, despite the relevance of the results, this study has some limitations. The sample was composed of disabled athletes. Typically, depending on the disability, such athletes depend less on the lower limbs to get around. This could explain why the athletes performed well even in the WW condition. Nevertheless, the current findings are still relevant for coaches and researchers for increased knowledge on warm-up and its effects on sport performance.

## 5. Conclusions

It was possible to conclude that the type of warm-up did not provide significant differences in the strength indicators. Thus, in powerlifting Paralympic, the type of warm-up does not seem to interfere with strength performance. Each type of warm-up may be at the discretion of each athlete that best suits him or her, or even there is the possibility of not doing any warm-up and also obtaining a positive result. Therefore, it is important to emphasize that although the types of applied warm-ups have not had a significant influence on the performance of muscle strength, it should be considered in order to preserve the athletes’ in relation to possible injuries.

## Figures and Tables

**Figure 1 medicina-56-00538-f001:**
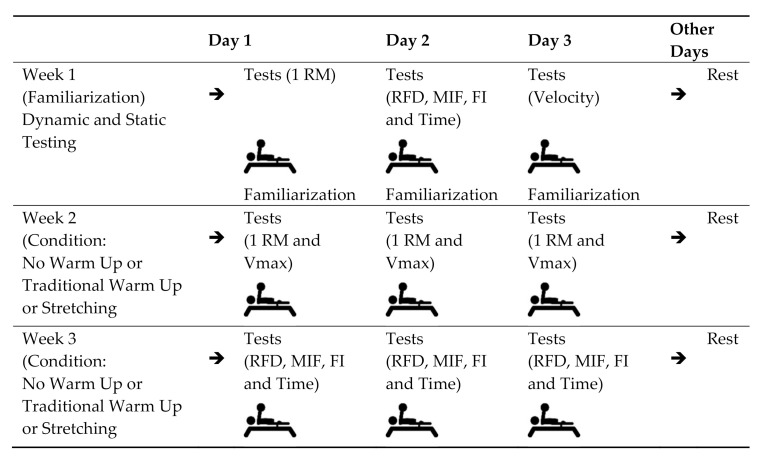
Experimental study design. MIF: maximum Isometric Force, RFD: rate of force development, FI: fatigue index, Tempo: time to maximum isometric force, Vmax: maximum speed.

**Figure 2 medicina-56-00538-f002:**
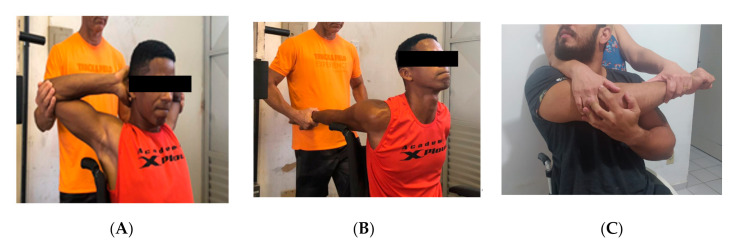
Stretches used as a warm-up method (**A**) pectoralis major, (**B**) shoulder and (**C**) triceps.

**Figure 3 medicina-56-00538-f003:**
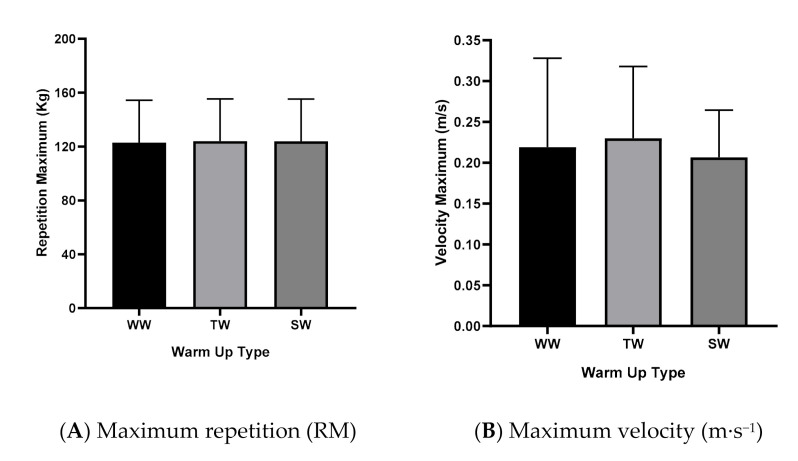
Evaluation of maximum repetition (RM) and maximum velocity (m·s^−1^), with different types of warm up. WW: without warm up, TW: traditional warm up and SW: stretching warm up.

**Figure 4 medicina-56-00538-f004:**
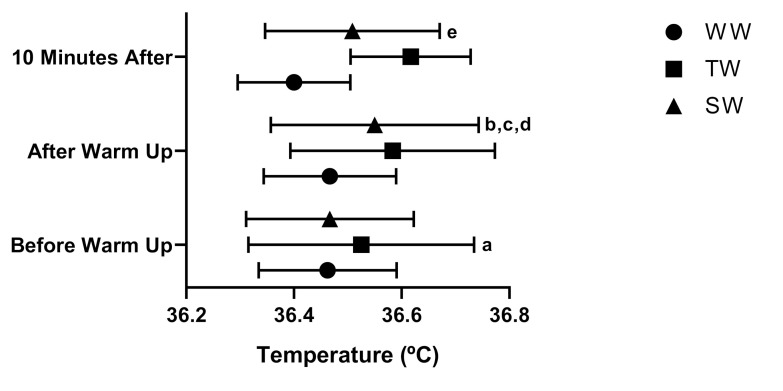
The evaluation of temperature (°C) in different types of warm up and at different times *p* < 0.05 (ANOVA two-way, and post hoc de Bonferroni), F = 26.87, η^2^_p_ = 0.710 (high effect). WW: without warm up, TW: traditional warm up and SW: stretching warm up. “a” *p* = 0.035; “b” *p* = 0.002; “c” *p* = 0.007; “d” *p* = 0.031; “e” *p* < 0.001.

**Table 1 medicina-56-00538-t001:** Sample characterization.

	Athletes
Age (years)	24.14 ± 6.21
Body weight (kg)	81.67 ± 17.36
Experience (years)	4.45 ± 0.31
1RM bench press test (kg)	126.25 ± 43.15
1RM/body weight	1.57 ± 0.34 *

* Values above 1.4 in the bench press, would be considered elite athletes, according to Ball and Wedman [20].

**Table 2 medicina-56-00538-t002:** Rate of force development (RFD) (N·s^−1^), maximum isometric force (MIF) (N), fatigue index (FI) (%) and time to maximum isometric force (Time) (s) (mean ± SD, 95% CI) in the different types of warm-up.

	RFD (N·m·s^−1^ s) X ± DP	MIF (N) X ± DP	FI (%) X ± DP	Time (s) X ± DP
	(IC 95%)	(IC 95%)	(IC 95%)	(IC 95%)
Without Warm up	4153.11 ± 1025.72 (1895.50–6410.71)	1005.07 ± 56.40 * (880.93–1129.21)	10.33 ± 1.25 (7.58–13.08)	2504.85 ± 311.39 (1819.48–3190.22)
Traditional	2810.93 ± 470.96 (1774.36–3847.50)	965.22 ± 56.23 (841.47–1088.98)	9.16 ± 0.83 (7.34–10.98)	2406.46 ± 430.78 (1458.33–3354.59)
Stretching	2522.67 ± 417.16 (1604.51–3440.83)	895.74 ± 45.62 * (795.33–996.15)	9.16 ± 1.03 (6.89–11.43)	2561.65 ± 419.72 (1637.86–3485.44)
*p*	0.476	0.005	0.940	0.999
η^2^_p_	0.136 ##	0.454 ###	0.055 #	0.006 #

* *p* < 0.05 (ANOVA two-way, and post hoc de Bonferroni). η^2^_p_: # low effect, ## medium effect, ### high effect. RFD: rate of force development; MIF: maximum isometric force, FI: fatigue index and time: time to maximum isometric force. * indicates difference in maximum isometric force (MIF), between without warm-up and stretching.

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
