# Peer review of "The Influence of Warm-Up on Body Temperature and Strength Performance in Brazilian National-Level Paralympic Powerlifting Athletes"

_medicina, 2020, doi:10.3390/medicina56100538_

Round 1

Reviewer 1 Report

The paperwork submitted for a review is interesting. It concerns a specific group of Paralympic athletes. Paralympic sport is developing dynamically, and the available literature rarely addresses the issue of training people with disabilities, therefore I highly appreciate the originality and innovation of the research presented in the paper.

Introduction:

  • The authors have introduced the issue by referring only to works concerning athletes without disabilities. In my opinion, it is reasonable to supplement the introduction with a review of research among disabled athletes. Physiological processes taking place in the body under the influence of warm-up at the cellular level in both groups (non-disabled and disabled) do not differ, but the specificity of warm-up varies. If there are no such publications, it is worth mentioning the originality and innovation of the authors' research.

Materials and Methods

  • Table 1 includes a variable called "1RM Bench Press Test (Kg)". It is unclear whether the results obtained in own research during the first week or the best result of a player in the last year were taken into account to calculate the average value.
  • In lines 97-98 the authors say: „In this study, we analyzed the effects of three different types of warm-up and (…)”.How is the effect of warm-up measured?
  • In line 98-100 the authors indicate that subsequent tests were performed with a 48-hour break. Figure 1 shows that each week the competitors performed particular tasks on consecutive days (Day 1, Day 2, Day 3). Such markings (Day 1, Day 2, Day 3) used in Figure 1 provide ambiguity.
  • In the subsection Temperature the authors wrote that the measurement was made before warm-um, after warm-up and 10 min after warm up. Please specify at what intervals the temperature in the WW (without warm-up) group was measured especially for the measurement "after warm up", if the warm-ups were not performed.

Results

  • The "*" used in Table 1 does not indicate what the authors have included in the text: „A significant difference was observed only for the MIF in the WW condition in relation to the TW and stretching conditions”.
  • In line 224-225 the authors state: „In addition, there was a significant difference between the WW in the “after” (between 36.4°C and 36.6°C) in relation to the SW in the “10 min later” condition”. And Figure 4 shows the variability among other parameters. Please verify.
  • In line 226-227 the authors wrote: “Finally, WW (36.3°C and 36.5°C) demonstrated a significant difference in relation to TW (between 36.5°C and 36.7°C) in the “10 min later” condition.” Please specify in this sentence: WW "before", "after" or "after 10 min". Such a marking is more clear than the use of the °C value in brackets.
  • In lines 225 and 227 the authors use the term "10 min later" and in Figure 4 "10 min after". Please standardize the nomenclature used.

Discussion

  • I suggest replacing "stretching methods" from line 229 with "warm-up methods". The authors use this phrase throughout the work.
  • In lines 230-235 the authors repeat the main results. Please consider removing this fragment. All the more so because, for example, in lines 243-244, the authors are repeating their results again but this time in relation to other authors' works.
  • In line 246 the authors use the abbreviation "FIM" instead of "MIF".

Conclusions

  • It should be emphasised more clearly in conclusion that the type of warm-up (or lack of warm-up) did not affect the test results in THIS study group. In my opinion, it seems important for the summary to include the value of warm-ups in order to prevent injuries.

Author Response

As attached document

Reviewer 2 Report

General Comments:

Your article covers an interesting topic and for the most part is well written and designed. However, I do have some specific recommendations that will help you improve the paper. Those are given below.

Specific recommendations:

 Abstract, Page 1

Line 30 – In this sentence you indicate that only “two” forms of warm-up were tested. Everywhere else you say there were “3” forms, using no warm-up (WW) as one of the trials. The athletes performed maximum isometric force (MIF), rate of force development (RFD), and speed test (Vmax) in two different methods of warm-up.

Line 38 – Here and elsewhere in the manuscript you talk about testing different types of heating. Granted pre-exercise warm-ups are designed to rase the temperature of muscles but the term “heating” suggests to the reader that you applied heat to the muscles prior to testing. Therefore, for the sake of clarity you need to eliminate the usage of “heating” and only use “warm-up” when discussing your forms of testing.

Introduction, Page 2

Line 47 – Add an “s” to “sport” at the end of this sentence. “Warm-up routines are a standard practice before training and competition in most sports.”

Line 58 – You need to clarify this sentence. I suggest the following changes. “Warming up can also help athletes in team sports by preparing them for the skills they need in basketball, handball, and baseball [9,10].

Methods and Materials

Page 2, line 81 – You need a “Subjects” heading for this section.

Page 3 Several aspects of your Experimental Design are confusing.

Line 98 – Here you indicate the following’ “Each participant completed the strength tests on the adapted bench press, after each warm-up condition in a random order (through a draw) separated by 48 h between each attempt.

Is the “draw” the same as the “lot method” you refer to on page 107? If so, only one of these descriptions should be used. 

Line 101 – Here you indicate that the study was performed over 3 weeks, with the first week aimed at familiarization of all of the tests to be performed, the second week dedicated to the 1RM and Vmax tests, and the third week to assess static force variables. You go on to say that in weeks 2 and 3 tests were assigned from three forms of heating (without warm-up [WW], traditional warm-up [TW], or stretching warm-up [SW]).

There are several problems here. First you need to define what you mean by “assigned”. I assume that it was the “lot method” but it is not clear. Secondly you need to change “heating” to “warm-up”.  Finally, you need to do a better job of clarifying what the “group” is in the last sentence.

The main problem is that you have two sections, Experimental Design on page 3 and Analytical Procedures and page 5, to describe what you did. All of this should be in the Experimental Design. It should be something like what I show below.    

Experimental Design  

In this study, we analyzed the effects of three different types of warm-up and the effect of each method on the strength of national-level PP athletes. All procedures were performed at the same time of day for and under the same environmental conditions (between 23°C and 25°C of temperature and relative humidity of ~60%). Following familiarization, each participant completed the strength tests on the adapted bench press [17]. The study was conducted over 3 weeks.

In week 1 the athletes all performed 3 familiarization sessions of the following tests: 1 maximum repetition (1RM), maximum isometric force (MIF) test with the bar at 15 cm from the chest, rate of force development (RFD), fatigue index (FI), time to maximum isometric force (time), and maximum speed (Vmax). In weeks 2 and 3, the athletes were randomly assigned, by drawing lots, to one of three warm-up forms; without warm-up [WW], traditional warm-up [TW], or stretching warm-up [SW]), with one third of the participants performing each type of warm-up in one of the evaluation sessions. At the beginning of testing in week 2 and 3, each participant remained seated for 5 min for the heart rate (Vantage NV; Polar, Kempele, Finland) and temperature measurements. After performing one of the three types of warm-up, the participants remained at rest for 10 min [28], and after that, they performed the force tests. The athletes were asked to maintain the same routine during the evaluation days, avoiding strenuous exercise and refraining from consuming caffeine 48 h before the test. The second week was dedicated to the 1RM and Vmax tests. The third week was dedicated to measuring static force variables. A summary of the design is presented in Figure 1.

Page 4, line 123 – Change “after heating” to “before warm-up” and “10 min after heating” to “10 min after heating” to “10 min after warm-up”. I am assuming that is what you measured.

Page 6

Line 191 – Change “types of heating” to “types of warm-up”.

Line 201 – Table 2 – You have several things here that need to be clarified. In the table description you have X+SD and 95% CI, but in the data section you have X+DP and IC 95%. Can you explain why? In addition, you have what I assume is the range of each measurement in parentheses below but do not indicate that anywhere. Finally, what does np2 stand for? I understand that this was defined somewhere in the text, but tables and figures need to stand alone so everything must be defined.

In closing I have two observations about your data.

The first one involves the RFD values of the WW warm-up. Based on the large difference between the WW and the TW and SW it would appear that the variation in the WW trial is why there was no significant differences. Although, it can’t be used in the discussion it does make you wonder why the TW and SW did not create the same variation.

The second one deals with the temperature data. It is interesting to note that the TW temperature was elevated well above WW and SW before the warm-up. Makes you wonder if their HR increased a little knowing they were going to perform a TW warm-up. Interesting data.

Author Response

As attached document

Reviewer 3 Report

GENERAL COMMENT: The authors should be acknowledged by their effort in conducting an applied research with elite level athletes. It is not easy to assess this type of athletic population which adds value to the present project. The study compared the effects of different warm-up protocols in Paralympic power-lifters and found that there were no meaningful differences between performing no warm-up, a traditional warm-up and a stretching warm-up. There are several aspects that must be improved before I can recommend the publication of the study. The introduction does not provide a sound rationale for the study. The discussion is too superficial and difficult to follow at times. However, I do believe that if the authors follow most of the recommendations herein the manuscript has potential to be published.

Please see specific comments below.

Abstract

The abstract will be address once the changes suggested to the manuscript are performed.

Introduction

In general, the introduction is well written and the authors must be acknowledged for it. However, I feel like there is a need to better  present the research question. My point is that the authors refer to the potential benefits of a proper warm-up but they do not mention the main characteristics of powerlifting. Moreover, it is not clear why there is the need to investigate the effects of these specific warm-up  protocols in power-lifters. What are the main warm-up practices in this sport? Is stretching common, for example?

I recommend the authors present the main characteristcs of power-lifting as a sport as well as the current state of the art when it comes to the warm-up protocols typically used in this population.

Specific comments:

Line 50. Please consider replacing "The" before "well-designed" by "A".

Line 63. "A bit separate" reads perhaps too colloquial. I suggest replacing this expression.

Line 64. Consider replacing "athlete performance" with "athletic performance".

Line 67. The term "post-activation" potentiation has been recently questioned. Blazevich and Babault (2019) make the case for using the term "post-activation performance enhancement". Please refer to the following reference for more on this topic and change the term accordingly: https://pubmed.ncbi.nlm.nih.gov/31736781/

Methods

Line 97-98. "In this study, we analyzed the effects of three different types of warm-up and the effect of this method on the strength of national-level PP athletes". This sentence in confusing. The authors state "the effects" twice which makes it hard to understand what is meant.

Line 105. What do the authors mean by "Static force variables"? Isometric tests? It is not clear for the reader.

Line 105-106. "In weeks 2 and 3, tests were assigned from three forms of heating". Please consider re-phrasing to: In weeks 2 and 3, tests were completed after each of the three forms of heating 

Line 123. Please consider replacing both "heating" by "the warm-up"

Line 125. "in which initially" seems lost on the sentence and makes it hard to understand what the authors mean. Please rephrase.

Line 127. Please consider replacing "lifted with one repetition" by "lifted once".

Line 135. In the RFD unit the -1 should be written in superscript.

Line 150. Why 5 s? Was there a specific reason for using that time duration? If the objective was to assess fatigue, wouldn't a longer sustained contraction potentially be more appropriate?

Line 151. What does "PT" stand for? If I am not mistaken, the authors do not detail what the abbreviation means.

Line 152. Why didn't the authors determine a fixed time interval for the calculation of the RFD. For example, 100 ms or 200 ms? Depending on the time interval used the results may change considerably. Please refer to https://pubmed.ncbi.nlm.nih.gov/26941023/ for more information on this.

Line 155. Analytical Procedures. What is described in the Analytical Procedures section is pretty much the same previously described in the Experimental Design section. Please consider removing the Analytical procedures section.

Line 180. Stretching Warm-up. Is this type of warm-up usually performed in this population? Do the authors consider it to have any ecological validity?

Line 193. Consider replacing "the size of the effect" with "effect size".

Results

Line 197. "time" for what?

Table 2. The units in the column of RFD (N.m.s-1sec) are not correct. Please amend.

Figure 4 is really difficult to understand. I suggest simplifying it for the reader.

Discussion

In general, the discussion needs substantial improvements. The text is not "fluid" and some of the ideas expressed are not clear and seem a bit out of place. A major revision of this section in particular is mandatory to bring the manuscript to a higher standard.

Specific comments:

Line 229. "stretching methods". Do the authors mean "warm-up protocols"?

Line 232. What is FDR?

Line 246. What does FIM mean? The authors use several abbreviations that are not detailed in the manuscript which makes it impossible to follow the ideas being presented.

Line 251-254. "Trajano et al. (...) indicators between warm-ups". What about immediately after? Why do the authors think that Trajano et al protocol yielded an acute drop in performance and the protocol used herein did not? Further discussion on this would be interesting and necessary.

Line 256-258. How are the findings by Janicijevic et al in line with the present study? Do Janicijevic and colleagues compare between warm-up protocols? The idea expressed by the authors is not clear to me.

Line 259-263. This entire paragraph seems "lost" within the context and line of the discussion. The authors must work on the fluidity of the text.

Author Response

As attached document

Round 2

Reviewer 3 Report

The authors have made the appropriate changes and have successfully addressed all my previous concerns. I consider the manuscript should be published in its current form. Congratulations to the authors.